# Efficiency of Sodium and Calcium Chloride in Conferring Cross-Tolerance to Water Deficit in Periwinkle

Nahid Zomorrodi [1], Abdolhossein Rezaei Nejad [1,*], Sadegh Mousavi-Fard [2], Hassan Feizi [3], Nikolaos Nikoloudakis [4] and Dimitrios Fanourakis [5]

1 Department of Horticultural Sciences, Faculty of Agriculture, Lorestan University, Khorramabad 68151-44316, Iran
2 Department of Horticultural Science, Faculty of Agriculture, Shahrekord University, Shahrekord P.O. Box 115, Iran
3 Department of Plant Production, Faculty of Agriculture, University of Torbat Heydarieh, Torbat Heydarieh 95161-68595, Iran
4 Department of Agricultural Sciences, Biotechnology and Food Science, Cyprus University of Technology, Limassol CY-3603, Cyprus
5 Laboratory of Quality and Safety of Agricultural Products, Landscape and Environment, Department of Agriculture, School of Agricultural Sciences, Hellenic Mediterranean University, Estavromenos, 71004 Heraklion, Greece
* Correspondence: rezaeinejad.h@lu.ac.ir

**Abstract:** The potential of using pre-stress NaCl or $CaCl_2$ applications to confer a cross-tolerance to a water deficit was evaluated in periwinkle. The plants initially received five applications of NaCl (0, 30 and 50 mM), or $CaCl_2$ (15 and 25 mM) via irrigation, and then they were cultivated under different water deficit regimes (80, 50 and 20% available water content). The water deficit induced smaller and denser stomata. It promoted a water use efficiency, a proline content and antioxidant enzyme activity. However, it downgraded the aesthetic value (plant stature, flower size and vegetation greenness), magnified the stem bending probability and strongly decreased the floral longevity. It additionally impeded the growth by reductions in the leaf area and photosynthesis. Plants undergoing a water deficit maintained a lower hydration and expressed oxidative damage symptoms, including enhanced chlorophyll and membrane degradation. As the water deficit intensified, these effects were more pronounced. Pre-stress $CaCl_2$ or NaCl applications generally restored most of the water severity-induced effects, with the former being more effective. For $CaCl_2$, the highest concentration (25 mM) was generally optimal, whereas NaCl was the lowest concentration (30 mM). In conclusion, pre-stress $CaCl_2$ or NaCl applications effectively confer a cross-tolerance to the water deficit by promoting the aesthetic value and extending the floral longevity, with the promotive effects being incremental as the water deficit becomes more severe.

**Keywords:** antioxidant defense; biomass accumulation; carbon assimilation; *Catharanthus roseus*; cellular damage; stomatal traits; water deprivation

## 1. Introduction

Across the world, a water deficit constraints the plant's yield and downgrades the product's quality [1]. Owing to on-going climate change, an upward trajectory in the severity of a water deficit is evident in large areas of the globe, which is evidently associated with progressively more adverse effects [2]. On this basis, a water deficit is a factor increasingly limiting the plant growth and productivity, and ways to ameliorate it are thus of great importance [1–3].

The exposure of plants to a specific stress condition has been shown to improve their tolerance to following stress events, which notably may be of a different nature compared to the preceding one [4,5]. In this perspective, the tolerance gained upon encountering a stress challenge alleviates the effects of another subsequent stress condition. This phenomenon

is termed cross-stress tolerance and it appears to be a promising way to improve the management of crop stress [6].

In this context, a previous exposure to osmotic stress (e.g., by using NaCl or CaCl$_2$) has been shown to confer a plant's tolerance to a water deficit [7–10]. The level of acquired cross-stress tolerance depends on the species, and it is concentration specific [7,8,11]. The beneficial role of NaCl and CaCl$_2$ has been, however, investigated in isolation, and in this way a comparative analysis is presently lacking. On this basis, which compound confers a better cross-stress tolerance currently remains unknown. Furthermore, previous reports mostly included one water deficit regime [7,11]. In this way, it remains elusive whether or not the compound and dose ought to be adjusted based on the severity of the water deficit.

Periwinkle [*Catharanthus roseus* (L.) G. Don] is an important ornamental as well as medicinal plant [1,12]. Despite its growing significance and multiple uses in floriculture and the pharmaceutical industry, studies focusing on the imposed stress and potential priming agents are scant. The aims of this study were (1) to assess the effects of NaCl and CaCl$_2$ on the plant growth and ornamental value under the control (non-stress) conditions, and (2) evaluate the comparative efficiency of these compounds as ameliorating agents when the plants are subjected to water deficits of various severities.

## 2. Materials and Methods

### 2.1. Plant Material and Growth Conditions

Seeds of a single lot of periwinkle cv. Pacifica XP Really Red were acquired from a commercial source (PanAmerican Seed Co., Chicago, IL, USA) and were selected based on the uniformity of their size. The seeds were sterilized (3 min) by an immersion in a sodium hypochlorite solution (3%; *v/v*), and then they were thoroughly washed with distilled water. Subsequently, they were sown in 2 L pots filled with a mixture of soil, sand (particle size of 0.08–2 mm) and composted cattle manure (1:1:1, *v/v*). Two seeds were manually placed at the center of each pot. At the cotyledon stage, the least vigorous seedling per pot was removed.

Before filling the pots, a growth media has been homogenized by sieving (6 mm). An equal weight of the growth media was employed to fill each pot. At the transplantation, the moisture content of the growth media was uniform within and among the pots. The growth media moisture content was computed by employing other pots (without plants) by considering the difference between saturated and dry conditions [3]. The saturation state was implemented by over-irrigation, followed by 48 h of recess, where the pots were enclosed in polyethylene bags. For evaluating the weight at the dry state, the growth media was placed in a drying oven (105 °C) for 24 h.

Subsequently, the potted plants were randomly arranged in a greenhouse compartment, located in the central-west part of Iran (Khorramabad, 33° N). A density of 20 plants m$^{-2}$ was applied. Based on a completely randomized design, fifteen experimental units (5 pre-stress treatments × 3 water deficit levels) were realized as a factorial experiment to evaluate whether a plant's exposure to osmotic stress (via NaCl or CaCl$_2$) enhances their tolerance to the subsequent water stress (the so-called cross-stress tolerance).

Following the adaptation (3 d), plants received an exogenous application of NaCl (0, 30 and 50 mM) or CaCl$_2$ (15 and 25 mM) via irrigation. The relevant concentration levels were determined based on a preliminary study (Figure S1). CaCl$_2$ (containing two Cl$^-$ ions) was employed in a half concentration of NaCl (containing one Cl$^-$ ion) to produce the same increase in the irrigation water Cl$^-$ ionic content and electrical conductivity. Five pre-stress applications were performed at 3 d intervals. At the first application, the plants were at the four-leaf stage. Following these applications, and for the two subsequent irrigation turns, the irrigation (10% run off) was performed by using double-distilled water to leash out the applied salts from the growth medium.

Following the above-mentioned (pre-stress) applications, three water deficit types were implemented via irrigation to an 80, 50 and 20% available water content. The available water content refers to the water that is available for the plant's absorption and is computed

by the difference in the field capacity and permanent wilting point. The field capacity and permanent wilting point were assessed via the retention curve method and refers to the water content at $-0.1$ and $-1.5$ kPa matric potentials, respectively. Each water deficit type was sustained by the daily regulation of the irrigation volume. Considering the daily irrigation routine and the sufficient growth medium volume (2 L), the day-to-day variation in the moisture content was most likely rather minimal [3].

The plants were cultivated under naturally fluctuating conditions of the temperature, relative air humidity and light. The average air temperature and relative air humidity were $22.2 \pm 1.6\,^{\circ}$C (range: 22–32 and 16–20 for day and night, respectively) and $61 \pm 1\%$ (range: 55–65), respectively. The mean daily light integral was $17.7 \pm 0.3$ mol m$^{-2}$ day$^{-1}$ (range: 15.2–21.6).

The measurements were undertaken at both the leaf and plant levels. For the leaf-scale assessments, the leaves under consideration had grown under direct light and were fully expanded. Replicate leaves were obtained from separate plants. The samples were acquired at the onset of the light period (06:00–07:00 h). The treatments were evaluated together. In the biomass determinations, the time between the sampling and the start of the evaluation was less than 10 min. In the remaining measurements, the samples were placed in vials, flash frozen in liquid nitrogen and transferred to a freezer ($-80\,^{\circ}$C). Sampling was conducted at the end of the cultivation period. In all the assessments, four replicates were considered per treatment.

### 2.2. Flower Bud Initiation, Opening and Longevity

The flower bud initiation, opening and longevity were assessed. The time to flowering (synonymous to time to visible bud) corresponds to the period between sowing and the appearance of flower buds (i.e., flower bud length $\approx 0.5$ cm). The longevity of an intact flower bud corresponds to the period between the opening and wilting (i.e., petal turgor loss).

### 2.3. Plant Growth, Morphology and Biomass Allocation

The lateral branch (number and length), main stem (length and diameter), flower (number and diameter), leaf (number and area), and root (volume and length) traits were determined. The main stem diameter was evaluated midway along its length, while the flower diameter corresponded to the mean of the largest diameter and the one perpendicular to it. For the determination of the leaf area (one-sided surface area), the leaves were scanned and assessed by using the Digimizer software (version 4.1.1.0, MedCalc Software, Ostend, Belgium) [2]. Before determining the root features, the pot was carefully inserted into a bucket filled with water for 1 h. After gently washing the growth medium, the root volume was assessed via the volume-displacement method [13]. The roots were immersed in a cylinder containing water. The volume of water dislocated by the roots corresponded to the root volume. The root diameter referred to the diameter of the largest sphere, which fitted into the root and including it [1]. The root length was considered as the distance between the shoot-to-root junction and the primary root tip.

The lateral branch, main stem, flower, leaf and root (fresh and dry) weight was also determined ($\pm 0.001$ g; Mettler ME303TE, Giessen, Germany). For evaluating the dry mass, the herbal material was dried in an oven for 72 h at 80 $^{\circ}$C [13]. By employing a dry weight, the specific leaf area (leaf area per leaf mass), flower mass ratio (flower mass per plant mass), leaf mass ratio (leaf mass per plant mass), root to shoot ratio (root mass per aboveground mass), stress tolerance index (dry weight per dry weight of the control; [14]) and water use efficiency (plant dry mass per unit of water consumed; [15]) were computed. The strength (mass per unit length) and tissue density (mass per unit volume) of the main stem were also determined, as indices of its sensitivity to bending [16,17].

### 2.4. Gas Exchange Traits

By using attached leaves, gas exchange traits were assessed in situ. Recordings were obtained via a portable photosynthesis system (CI-340; CID, Inc., Camas, WA, USA). The measurement cuvette (6.25 cm$^2$) was adjusted to a 22 °C air temperature, a 50% relative air humidity, a 400 μmol mol$^{-1}$ incoming $CO_2$ concentration and a 200 μmol m$^{-2}$ s$^{-1}$ light intensity [18]. To ensure the steady state stomatal conductance, measurements were undertaken 2 h after the onset of the light period [18,19].

### 2.5. Chlorophyll and Carotenoid Contents

The leaf chlorophyll content is elemental for a carbon assimilation, while carotenoids are critical non-enzymatic antioxidants [20,21]. Following slicing, the samples (0.1 g) were homogenized by adding 10 mL of 100% acetone. The extract was subsequently centrifuged (14,000× $g$ for 20 min) and the supernatant was obtained. Given the light sensitivity of chlorophyll, the extraction took place under darkness [22,23]. The extract was evaluated by using a spectrophotometer (Mapada UV-1800; Shanghai Mapada Instruments Co., Ltd., Shanghai, China). The chlorophyll and carotenoid contents were computed as described by Lichtenthaler and Wellburn (1983) [24].

### 2.6. Stomatal and Epidermal Cell Anatomical Characteristics

The leaf stomatal and epidermal cell characteristics were evaluated. The sampling area was selected between the base and apex, as well as the halfway main vein and lateral margin [25,26]. The abaxial leaf side was covered with a thin layer of nail polish. Following the drying (10 min), the polish was removed by a transparent tape and was fixed on a microscopic slide. Determinations were established by using a camera-attached light microscope. Images were obtained through the Omax software (ver. 3.2, Omax Corp., Kent, WA, USA). The per replicate leaf, stomatal dimensions were assessed on 10 stomata (magnification of 100×), while the stomatal and epidermal cell densities were evaluated on five fields of view (magnification of 10×) [27]. The stomatal index [i.e., stomatal number per stomatal and epidermal cell number] was calculated [27]. The stomatal size corresponded to the stomatal length multiplied by the stomatal width [17,27]. The stomatal area per leaf area was calculated (stomatal size × stomatal density). Subsequently, the epidermal cell area per leaf area was calculated (10$^6$–stomatal area per leaf area) [1]. The mean epidermal cell size was calculated by dividing the epidermal cell area per leaf area by the epidermal cell density [1]. Image processing was applied with ImageJ software (Wayne Rasband/NIH, Bethesda, MD, USA).

### 2.7. Water Status

The leaf water status was evaluated by determining the relative water content (RWC). Sampling was conducted 3 h after the onset of the photoperiod [26]. Following the leaf excision, the fresh weight was gravimetrically determined (±0.0001 g; Mettler AE 200, Giessen, Germany). Then, the samples were floated on double-distilled water inside a Petri dish and covered with a lid. Following the incubation (24 h), the obtained weight was considered as the turgid (saturated) one. Then, the dry weight (72 h at 80 °C) was obtained. The RWC was computed as described by Taheri-Garavand et al. (2021) [19].

### 2.8. Proline Content

Proline contributes to the cell osmotic regulation by means of a water potential decrease, and in this perspective protects both the enzyme activity and macromolecules' structure [14]. On this basis, the effects of the treatment on the proline content were evaluated. The leaf samples (0.5 g) were homogenized and then inserted in 10 mL of 3% (*w/v*) aqueous sulphosalycylic acid. The extract was filtered (Whatmann No. 2 filter paper) and 2 mL of the filtrate were combined with 2 mL of acid–ninhydrin and 2 mL of glacial acetic acid. The resultant solution was heated (100 °C for 1 h). The reaction mixture was extracted with 4 mL of toluene, and the chromophore containing toluene was aspirated from the

liquid phase. After equilibration at 25 °C, the absorbance was evaluated at 520 nm by using a spectrometer (Mapada UV-1800, Shanghai. Mapada Instruments Co., Ltd., Shanghai, China). The proline concentration was computed by using a calibration curve [3].

### 2.9. Electrolyte Leakage

The relative ion content in the apoplastic space, taken as an index of the membrane's stability, was assessed by determining the electrolyte leakage [22,28]. The leaf discs (1 cm$^2$) were washed 3 times (3 min) with double-distilled water (to take away surface-adhered electrolytes) and were then floated on 10 mL of double-distilled water. The vials were then shaken (3 min). After 24 h of floating at room temperature (25 °C), the electrolyte leakage in the solution was assessed by using a conductimeter (Crison 522, Crison Instruments, S.A., Spain). Subsequently, the samples were autoclaved (120 °C for 20 min) and the total conductivity was acquired after equilibration at room temperature (25 °C). The results are shown as a percentage of the total conductivity. Per replicate leaf, four discs were evaluated.

### 2.10. Lipid Peroxidation

The malondialdehyde (MDA) content, considered as an index of the lipid peroxidation level, was assessed by using the thiobarbituric acid reactive substance assay [22,28]. The leaf discs (1 cm$^2$) were homogenized and then inserted in 5 mL of 20% ($w/v$) trichloroacetic acid and 0.5% ($w/v$) thiobarbituric acid. Then, the suspension was centrifuged ($6000\times g$ for 15 min). The resultant solution was heated (100 °C for 25 min). After equilibration at room temperature (25 °C), the precipitate was removed through centrifugation ($6000\times g$ for 5 min). The MDA content was computed from the absorbance at 532 nm after deducting the non-specific absorption at 450 and 600 nm (Mapada UV-1800; Shanghai Mapada Instruments Co., Ltd., Shanghai, China). The extinction coefficient of 156 mmol MDA L$^{-1}$ cm$^{-1}$ was employed [23,28]. Per replicate leaf, four discs were evaluated.

### 2.11. Enzymatic Activity

The catalase (CAT) activity was assessed as described by Ahmadi-Majd et al. (2022) [28]. The leaf segments (1 cm$^2$) were ground in liquid nitrogen, homogenized with 1.5 mL of potassium phosphate buffer (containing 1 mM of EDTA and 2% polyvinylpyrrolidone) and centrifuged ($14000\times g$ for 20 min) at 4 °C. The CAT activity in the supernatant was assessed by following the decrease in the absorbance at 240 nm for 2 min (10 s intervals) in a reaction mixture containing a potassium phosphate buffer and H$_2$O$_2$. The extinction coefficient of 39.4 M$^{-1}$ cm$^{-1}$ was used. The CAT activity was expressed as µmol of H$_2$O$_2$ reduced min$^{-1}$ g$^{-1}$ tissue.

The peroxidase (POD) activity was assayed as described by Ahmadi-Majd et al. (2022) [28]. The leaf segments (1 cm$^2$) were ground in liquid nitrogen, homogenized with 1.5 mL of 50 mM potassium phosphate buffer (pH 7.0) and centrifuged ($14000\times g$ for 20 min) at 4 °C. The POD activity in the supernatant was assessed by following the decrease in the absorbance at 470 nm for 2 min (10 s intervals) in a reaction mixture containing a potassium phosphate buffer, guaiacol and H$_2$O$_2$. The extinction coefficient of 26.6 mM$^{-1}$ cm$^{-1}$ was used. The POD activity was expressed as µmol of H$_2$O$_2$ reduced min$^{-1}$ g$^{-1}$ tissue.

### 2.12. Statistical Analysis

The data were subjected to an analysis of variance by using SPSS 23 (SPSS Inc., Chicago, IL, USA). A two-way ANOVA was employed, with the water deficit level as the main factor and the pre-stress treatment as the split factor. The data were first tested for normality (Shapiro–Wilk test) and homogeneity of variances (Levene's test). Subsequently, the estimated least significant differences (LSD) of the treatment effects were determined ($p = 0.05$).

For the 15 experimental units, the eigenvalues were extracted and the most contributing variables for each dimension were computed and identified. The first two eigenvalues cumulated 62.8% of the total variance and were retained to produce the principal components. A principal component analysis (PCA) was produced to depict correlations across

the water deficit levels and spray treatments to the principal components. The individuals were grouped (by discrete color) and variable by their contribution to the principal components (gradient colors). A correlation plot was also computed in order to depict positive and negative associations across the variables inquired. The "corrplot", "FactoMineR", "factoextra" and "readxl" libraries were used under the R-studio integrated development environment (RStudio suite V 1.2.5033).

## 3. Results

### 3.1. Gas Exchange Traits

As the water deficit intensified (i.e., the substrate available water content was held lower), the four gas exchange traits under study (the transpiration rate, stomatal conductance, internal $CO_2$ concentration and photosynthesis rate) decreased (Figure 1). Independently of the water deficit severity, pre-stress applications generally enhanced the gas exchange traits. This enhancement was consistently more pronounced via $CaCl_2$ compared to NaCl. For $CaCl_2$, the highest employed concentration (25 mM) was generally more efficient in promoting the above-noted positive effects compared to the lowest one (15 mM). For NaCl, the optimal concentration was dependent on the water deficit level. For instance, the lowest one (30 mM) was the most efficient in control (non-stress) conditions (80% available water content), whereas the highest one (50 mM) was most efficient at a 50% available water content.

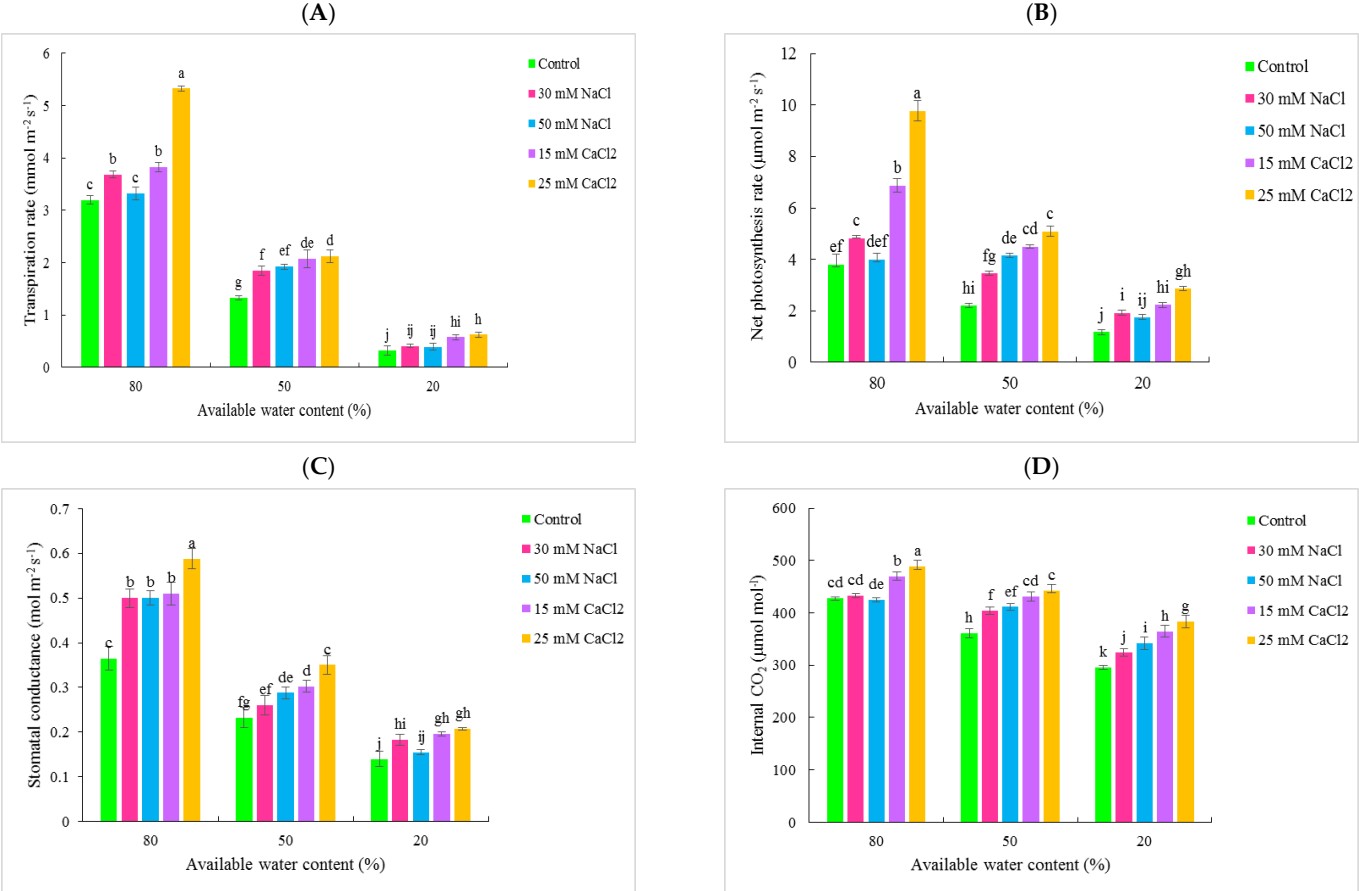

**Figure 1.** Transpiration (**A**), photosynthesis (**B**), stomatal conductance (**C**) and internal $CO_2$ concentration (**D**) of periwinkle cv. 'Pacifica XP Really Red' plants initially receiving NaCl (0, 30 and 50 mM) or $CaCl_2$ (15 and 25 mM) via irrigation (five applications at 3 d intervals), and subsequently exposed to different watering levels (80, 50 and 20% available water content) during cultivation. At first application, plants were at the four-leaf stage. Values with different letters indicate significant differences (*p* = 0.05; comparison in columns). Values are the mean of four replications.

When pooling all 15 treatments, the stomatal conductance was positively correlated with the transpiration rate, internal $CO_2$ concentration and photosynthesis rate ($R^2$ of 0.81, 0.51 and 0.71, respectively; see also Figure 2). The internal $CO_2$ concentration and photosynthesis rate were also positively correlated ($R^2 = 0.70$).

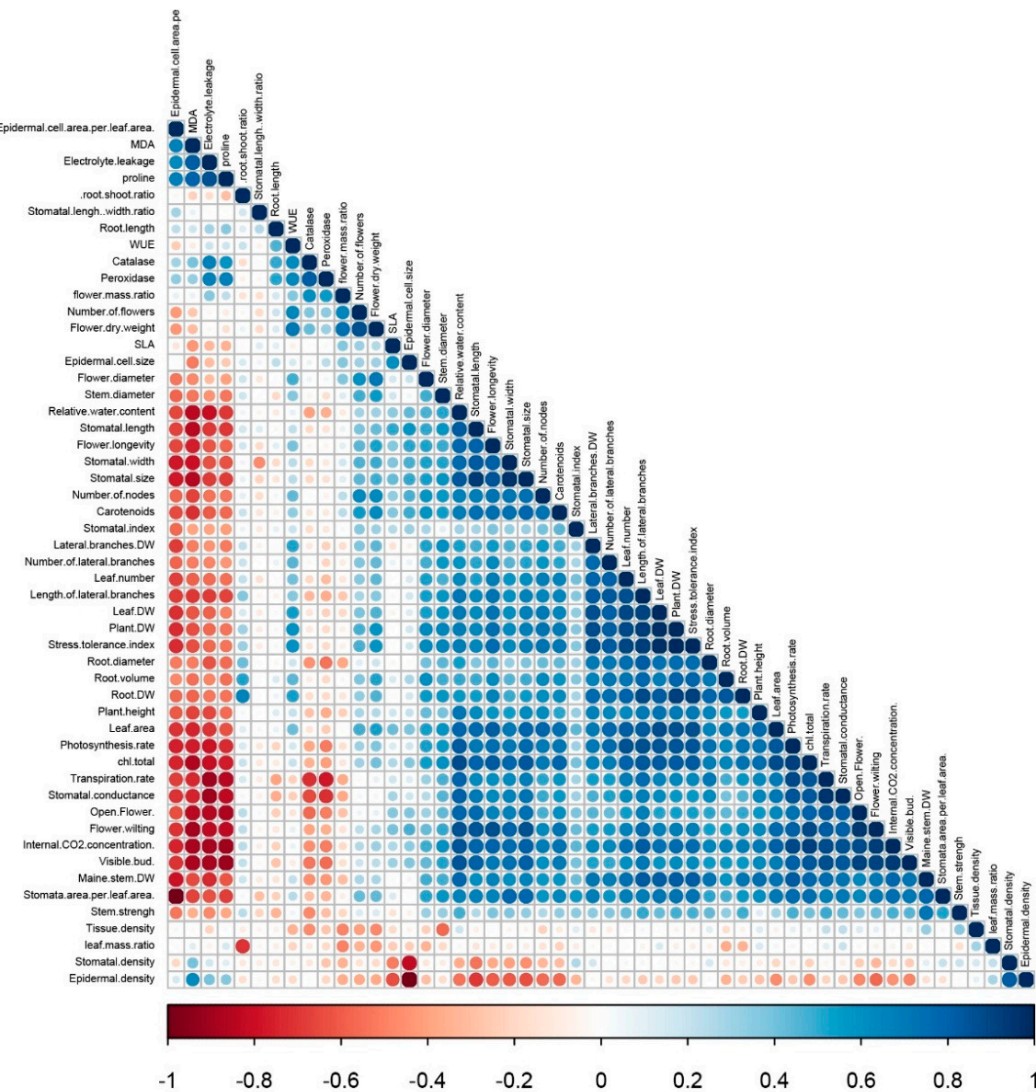

**Figure 2.** Positive, neutral and negative affinities across morpho-physiological traits in periwinkle cv. 'Pacifica XP Really Red'. Plants initially received NaCl (0, 30 and 50 mM) or $CaCl_2$ (15 and 25 mM) via irrigation (five applications at 3 d intervals), and subsequently exposed to different watering levels (80, 50 and 20% available water content) during cultivation. At first application, plants were at the four-leaf stage. Positive correlations are indicated by blue circles, while negative associations are expressed by red circles. The intensity of color corresponds to the correlation coefficient (r) ranging from −1 to 1 (scale). The larger size of circles indicates statistically significant values (non-significant, $p = 0.05$, $p = 0.01$ and $p = 0.001$, respectively).

### 3.2. Flower Bud Initiation, Opening and Longevity

As the water deficit intensified, the periods for the flower bud initiation, opening and longevity were all decreased (Table 1). When comparing the effect of the water deficit in these periods, the noted decrease was much stronger in the flower buds' longevity. Independently of the water deficit severity, the pre-stress applications generally enhanced the periods for the flower buds' initiation, opening and longevity. For the flower buds' longevity, this enhancement was generally more prominent via $CaCl_2$ compared to NaCl. For $CaCl_2$, the highest employed concentration (25 mM) was generally more effective in

promoting the flower buds' longevity compared to the lowest one (15 mM). For NaCl, no significant difference was noted in the longevity of the flower buds among the two employed concentrations (30 and 50 mM).

**Table 1.** Time to flowering (also referred as time to visible bud), time to open flower, time to wilting (i.e., petal turgor loss) and intact (on plant) flower bud longevity [(time to wilting)—(time to open flower)] of periwinkle cv. 'Pacifica XP Really Red' plants initially receiving NaCl (0, 30 and 50 mM) or $CaCl_2$ (15 and 25 mM) via irrigation (five applications at 3 d intervals), and subsequently exposed to different watering levels (80, 50 and 20% available water content) during cultivation. At first application, plants were at the four-leaf stage. Means followed by different letters indicate significant differences ($p = 0.05$; comparison in columns). Values are the mean of four replications.

| Treatment | | | Time to | | | Intact Flower Bud Longevity (d) |
|---|---|---|---|---|---|---|
| Available Water Content (%) | Compound | Concentration (mM) | Visible Bud (d) | Open Flower (d) | Flower Wilting (d) | |
| 80 | - | 0 | 74.2 bc | 90.2 b | 97.5 de | 7.2 f |
| | NaCl | 30 | 74.7 b | 92.5 ab | 103.2 c | 10.7 de |
| | | 50 | 74.2 bc | 93.0 a | 102.5 c | 9.5 e |
| | $CaCl_2$ | 15 | 77.2 a | 92.2 ab | 106.2 b | 14.0 ab |
| | | 25 | 78.2 a | 93.7 a | 109.5 a | 15.7 a |
| 50 | - | 0 | 69.0 fg | 84.5 d | 88.0 g | 3.5 g |
| | NaCl | 30 | 71.2 de | 86.2 cd | 97.4 de | 11.1 cde |
| | | 50 | 72.7 cd | 87.5 c | 98.0 de | 10.5 de |
| | $CaCl_2$ | 15 | 70.7 ef | 86.2 cd | 96.5 ef | 10.2 de |
| | | 25 | 74.5 bc | 86.5 cd | 99.2 d | 12.7 bc |
| 20 | - | 0 | 58.2 j | 72.7 g | 74.7 i | 2.0 g |
| | NaCl | 30 | 64.2 i | 76.7 f | 82.7 h | 6.0 f |
| | | 50 | 67.5 gh | 81.0 e | 88.2 g | 7.2 f |
| | $CaCl_2$ | 15 | 66.7 h | 84.7 d | 94.7 f | 10.0 de |
| | | 25 | 67.0 h | 86.0 cd | 97.7 de | 11.7 cd |

*3.3. Plant Growth, Morphology, and Biomass Allocation*

A range of morphological traits were evaluated involving various organs (i.e., the lateral branches, main stem, leaves, flower and root; Table 2). As the water deficit intensified, the traits under study generally underwent lower values, besides the number of lateral branches and root length (Table 2). Under the water deficit conditions (50 and 20% available water content), the pre-stress NaCl application generally improved the above-noted morphological traits across the organs (Figures S2 and S3), while for the application of $CaCl_2$, this enhancement was independent of the water deficit severity (80, 50 and 20% available water content). This effect was generally more prominent via $CaCl_2$ compared to NaCl. For $CaCl_2$, the differences among the two concentrations (15 and 25 mM) were often non-significant, besides some traits where the highest concentration (25 mM) was more efficient than the lowest one (15 mM). For NaCl, the differences among the two concentrations (15 and 25 mM) were non-significant in some traits, while in others the lowest concentration (30 mM) was generally optimal compared to the highest one (50 mM).

As the water deficit intensified, both the stem bending sensitivity indices (the strength and tissue density) shifted to lower values (Table 2). In some cases, the pre-stress applications caused a decrease in the stem bending sensitivity indices, while in most cases, the effect was not significant.

**Table 2.** Morphological features of periwinkle cv. 'Pacifica XP Really Red' plants initially receiving NaCl (0, 30 and 50 mM) or CaCl$_2$ (15 and 25 mM) via irrigation (five applications at 3 d intervals), and subsequently exposed to different watering levels (80, 50 and 20% available water content) during cultivation. At first application, plants were at the four-leaf stage. Means followed by different letters indicate significant differences ($p$ = 0.05; comparison in columns). Values are the mean of four replications.

| Treatment | | | Lateral Branches | | | Main Stem | | | Number of Nodes | Leaf | | Flower | | | Root | |
| --- | --- | --- | --- | --- | --- | --- | --- | --- | --- | --- | --- | --- | --- | --- | --- | --- |
| Available Water Content (%) | Compound | Concentration (mM) | Number | Length (cm) | Length (cm) | Diameter (mm) | Strength (g cm$^{-1}$) | Tissue Density (g cm$^{-3}$) | | Number | Area (cm$^2$) | Number | Diameter (mm) | Length (cm) | Diameter (mm) | Volume (cm$^3$) |
| 80 | - | 0 | 8.0 de | 9.2 cd | 9.7 bcd | 3.40 cde | 18.9 a | 2.09 a | 9.0 de | 67.5 ef | 157 h | 3.5 fg | 33.4 c–f | 20.1 ef | 3.24 bc | 4.25 bc |
| | NaCl | 30 | 8.3 d | 7.6 efg | 10.1 bc | 3.21 efg | 14.3 bcd | 1.77 abc | 10.0 bc | 75.5 cd | 234 d | 5.5 b–e | 29.9 fgh | 22.0 cde | 2.84 c–f | 3.50 cd |
| | | 50 | 8.0 de | 6.1 h | 9.5 cd | 3.63 bc | 16.0 b | 1.56 bcd | 8.8 e | 53.5 hi | 180 g | 3.2 fg | 28.7 gh | 18.7 f | 2.50 fgh | 2.00 f |
| | CaCl$_2$ | 15 | 12.8 a | 11.2 b | 12.7 a | 3.34 def | 18.7 a | 2.16 a | 10.5 ab | 87.3 b | 273 b | 4.0 efg | 32.7 c–g | 24.6 ab | 3.62 ab | 5.25 a |
| | | 25 | 13.8 a | 14.6 a | 12.0 a | 3.91 a | 19.7 a | 1.67 bcd | 11.3 a | 106.3 a | 307 a | 7.0 b | 39.5 a | 20.2 ef | 3.95 a | 5.50 a |
| 50 | - | 0 | 8.5 cd | 6.1 h | 9.2 cde | 3.29 def | 12.0 e | 1.41 cd | 7.8 f | 63.8 fg | 131 i | 3.0 g | 30.2 e–h | 22.0 cde | 2.89 c–f | 3.50 cd |
| | NaCl | 30 | 9.0 cd | 7.2 fgh | 9.5 cd | 3.54 bcd | 12.4 de | 1.27 de | 9.5 cde | 65.5 fg | 203 f | 6.2 bcd | 35.6 abc | 23.8 abc | 2.82 d–g | 5.00 ab |
| | | 50 | 8.3 cd | 6.3 gh | 8.1 ef | 3.44 cde | 12.3 de | 1.34 d | 10.0 bc | 72.5 de | 148 h | 6.0 bcd | 34.9 bcd | 20.7 de | 2.25 hi | 3.50 cd |
| | CaCl$_2$ | 15 | 11.0 b | 10.1 bc | 10.0 bc | 3.66 abc | 9.5 f | 0.91 e | 10.5 ab | 81.3 bc | 250 c | 6.2 bcd | 34.7 b–e | 23.0 bc | 3.00 cde | 5.75 a |
| | | 25 | 11.0 b | 8.8 cde | 10.7 b | 3.75 ab | 15.0 bc | 1.37 d | 10.5 ab | 75.5 cd | 237 d | 10.2 a | 38.6 ab | 22.0 cde | 3.02 cd | 5.00 ab |
| 20 | - | 0 | 7.5 de | 3.6 i | 7.7 f | 3.02 g | 13.2 cde | 1.85 ab | 7.3 f | 57.5 hi | 90 k | 2.7 g | 25.9 h | 22.2 cd | 2.52 fgh | 1.75 f |
| | NaCl | 30 | 8.3 d | 4.3 i | 9.2 cde | 3.33 def | 12.7 de | 1.47 bcd | 9.8 bcd | 59.5 gh | 183 g | 6.7 bc | 32.3 c–g | 22.5 cd | 2.44 gh | 3.00 de |
| | | 50 | 6.5 e | 4.2 i | 8.6 def | 3.10 fg | 12.4 de | 1.65 bcd | 8.8 e | 53.0 i | 116 j | 4.7 def | 30.5 d–g | 23.1 bc | 1.92 i | 2.13 ef |
| | CaCl$_2$ | 15 | 8.5 cd | 6.3 gh | 9.0 cde | 3.19 efg | 13.5 cde | 1.78 abc | 8.8 e | 64.0 fg | 157 h | 5.2 cde | 30.4 d–g | 24.5 ab | 2.62 e–h | 3.63 cd |
| | | 25 | 10.0 bc | 8.5 def | 9.7 bcd | 3.52 bcd | 14.0 b–e | 1.44 cd | 9.8 bcd | 71.8 de | 219 e | 7.0 b | 34.7 bcd | 25.1 a | 2.56 fgh | 3.25 d |

As the water deficit intensified, the water use efficiency (the plant dry mass per unit of water absorbed) increased, and the stress tolerance index (the dry weight relative to the dry weight of the control) decreased (Figure 3). The plants' dry weight also became lighter owing to the lower individual organ weight (Table 3). This was evident in all organs (the main stem, lateral branches, leaves and root), besides the flower. Independently of the water deficit severity, the pre-stress $CaCl_2$ application generally improved the water use efficiency, stress tolerance index and biomass accumulation, especially at the highest concentration (25 mM). Under the control (non-stress) conditions, the pre-stress NaCl application generally caused an adverse effect on the water use efficiency, stress tolerance index and biomass accumulation, whereas under water deficit conditions it induced a positive effect, especially at the lowest concentration (30 mM).

**(A)**

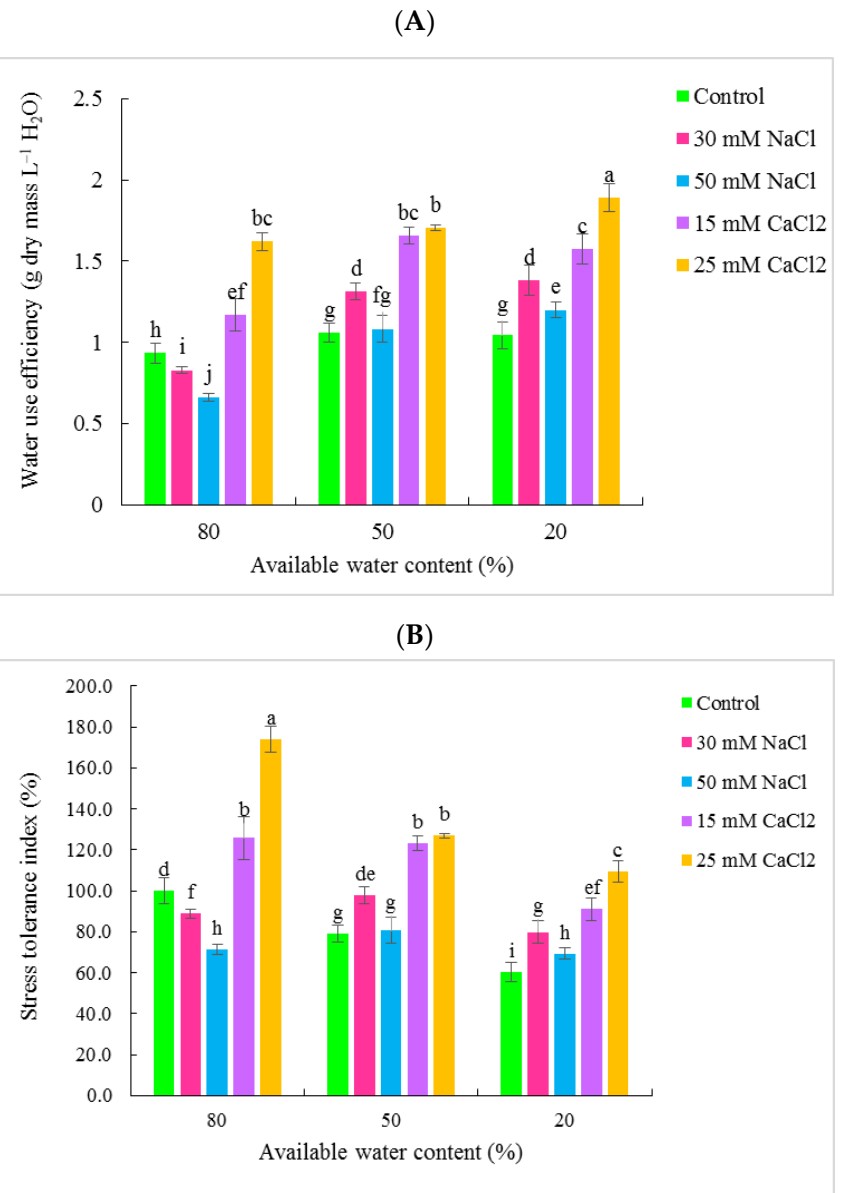

**(B)**

**Figure 3.** Water use efficiency (**A**) and stress tolerance index (**B**) of periwinkle cv. 'Pacifica XP Really Red' plants initially receiving NaCl (0, 30 and 50 mM) or $CaCl_2$ (15 and 25 mM) via irrigation (five applications at 3 d intervals), and subsequently exposed to different watering levels (80, 50 and 20% available water content) during cultivation. At first application, plants were at the four-leaf stage. Values with different letters indicate significant differences ($p = 0.05$; comparison in columns). Values are the mean of four replications.

**Table 3.** Plant growth, biomass allocation and morphology of periwinkle cv. 'Pacifica XP Really Red' plants initially receiving NaCl (0, 30 and 50 mM) or CaCl$_2$ (15 and 25 mM) via irrigation (five applications at 3 d intervals), and subsequently exposed to different watering levels (80, 50 and 20% available water content) during cultivation. At first application, plants were at the four-leaf stage. Means followed by different letters indicate significant differences (*p* = 0.05; comparison in columns). Values are the mean of four replications. STI, stress tolerance index, WUE, water use efficiency.

| Available Water content (%) | Treatment Compound | Concentration (mM) | Main Stem | Lateral Branches | Leaf | Flower | Root | Plant | Root to Shoot Ratio | Specific Leaf Area (cm² g⁻¹) | Leaf Mass Ratio | Flower Mass Ratio |
|---|---|---|---|---|---|---|---|---|---|---|---|---|
| | | | | | Dry Weight (g) | | | | | | | |
| 80 | - | 0 | 0.17 c | 0.03 cd | 0.59 ef | 0.06 hi | 0.34 bc | 1.21 d | 39.3 a | 264 c | 0.49 ef | 0.06 g |
| | NaCl | 30 | 0.13 de | 0.01 ef | 0.60 e | 0.10 fg | 0.21 ef | 1.07 f | 24.6 efg | 385 a | 0.56 ab | 0.09 cd |
| | | 50 | 0.13 def | 0.01 ef | 0.48 hi | 0.06 hi | 0.16 fg | 0.86 h | 24.2 fg | 373 a | 0.56 ab | 0.07 ef |
| | CaCl$_2$ | 15 | 0.18 b | 0.04 bc | 0.85 b | 0.08 gh | 0.35 b | 1.52 b | 30.6 c–f | 322 b | 0.56 ab | 0.06 g |
| | | 25 | 0.20 a | 0.07 a | 1.13 a | 0.18 b | 0.50 a | 2.10 a | 31.6 b–e | 270 c | 0.54 bc | 0.09 de |
| 50 | - | 0 | 0.11 gh | 0.03 bc | 0.50 hi | 0.06 hi | 0.23 de | 0.95 g | 32.6 a–d | 262 c | 0.52 cde | 0.07 fg |
| | NaCl | 30 | 0.12 efg | 0.02 def | 0.53 gh | 0.17 bc | 0.33 bc | 1.18 de | 38.8 ab | 317 a | 0.45 g | 0.15 a |
| | | 50 | 0.11 gh | 0.02 def | 0.47 ij | 0.14 de | 0.22 ef | 0.97 g | 29.3 c–g | 317 b | 0.48 f | 0.15 a |
| | CaCl$_2$ | 15 | 0.12 fg | 0.04 b | 0.79 c | 0.16 cd | 0.36 b | 1.49 b | 32.4 a–d | 314 b | 0.53 bcd | 0.11 bc |
| | | 25 | 0.17 bc | 0.04 b | 0.75 cd | 0.22 a | 0.33 bc | 1.53 b | 27.6 d–g | 312 b | 0.49 ef | 0.15 a |
| 20 | - | 0 | 0.10 i | 0.01 ef | 0.42 j | 0.05 i | 0.13 g | 0.73 i | 22.9 g | 205 d | 0.58 a | 0.08 ef |
| | NaCl | 30 | 0.11 gh | 0.01 ef | 0.49 hi | 0.14 de | 0.19 efg | 0.96 g | 25.3 d–g | 367 a | 0.52 c–f | 0.15 a |
| | | 50 | 0.10 hi | 0.01 f | 0.45 ij | 0.10 f | 0.16 fg | 0.84 h | 24.3 efg | 255 c | 0.54 bc | 0.12 b |
| | CaCl$_2$ | 15 | 0.12 fg | 0.02 de | 0.55 fg | 0.11 f | 0.28 cd | 1.10 ef | 35.1 abc | 310 b | 0.51 def | 0.10 cd |
| | | 25 | 0.13 d | 0.04 bc | 0.72 d | 0.13 e | 0.28 cd | 1.32 c | 27.8 c–g | 326 b | 0.55 bc | 0.10 cd |

When pooling all the 15 treatments, the plants' dry weight was positively correlated with the dry weight of the main stem, lateral branches, leaf, flower and root ($R^2$ of 0.91, 0.96, 0.97, 0.97 and 0.95, respectively; see also Figure 2). A larger flower mass was positively associated with a larger flower diameter ($R^2 = 0.61$; see also Figure 2). The roots' dry mass was positively related to the roots' diameter and volume ($R^2$ of 0.82 and 0.91, respectively), but not with the roots' length ($R^2$ of 0.00; see also Figure 2).

As the water deficit intensified, the root to shoot ratio and specific leaf area (indicative of the leaves' thickness) tended to decrease, whereas the leaf and flower mass ratios followed the opposite trend (Table 3). In the root to shoot and leaf mass ratios, the effect of pre-stress applications was dependent on the water deficit level. Independently of the water deficit severity, pre-stress applications generally promoted the specific leaf area and flower mass ratio.

### 3.4. Stomatal and Epidermal Cell Anatomical Characteristics

The leaf area was mostly covered by epidermal cells (94.9–97.5%), while stomata acquired only a small portion of the epidermis (2.48–5.06%; Table 4). As the water deficit intensified, the stomatal size and index decreased, whereas the stomatal and epidermal cell densities increased. Limited effects were noted on the stomatal length to width ratio, indicating that the stomatal shape was little affected. The pre-stress applications generally increased the stomatal size and decreased the stomatal and epidermal cell densities.

**Table 4.** Epidermal cell and stomatal anatomical features of periwinkle cv. 'Pacifica XP Really Red' plants initially receiving NaCl (0, 30 and 50 mM) or $CaCl_2$ (15 and 25 mM) via irrigation (five applications at 3 d intervals), and subsequently exposed to different watering levels (80, 50 and 20% available water content) during cultivation. At first application, plants were at the four-leaf stage. Means followed by different letters indicate significant differences ($p = 0.05$; comparison in columns). Values are the mean of four replications.

| Treatment | | | Epidermal Cell | | | Stomatal | | | | | | |
|---|---|---|---|---|---|---|---|---|---|---|---|---|
| Available Water Content (%) | Compound | Concentration (mM) | Density (mm$^{-2}$) | Size (μm$^2$) | Area Per Leaf Area (%) | Density (mm$^2$) | Index (%) | Length (μm) | Width (μm) | Size (μm$^2$) | Length to Width Ratio | Area Per Leaf Area (%) |
| 80 | - | 0 | 575 cd | 1666 ef | 95.8 fg | 155 d | 21.2 cd | 26.5 gh | 10.1 fgh | 536 gh | 2.65 a–d | 4.16 cd |
| | NaCl | 30 | 500 h | 1911 a | 95.4 h | 145 e | 22.5 ab | 28.3 b–e | 11.1 bc | 632 b | 2.54 bcd | 4.59 b |
| | | 50 | 572 cd | 1677 ef | 95.9 f | 135 f | 19.1 fg | 28.7 bc | 10.6 c–f | 609 bcd | 2.71 ab | 4.12 d |
| | CaCl$_2$ | 15 | 510 gh | 1878 ab | 95.7 fgh | 137 f | 21.2 cd | 29.0 b | 10.7 cde | 621 bc | 2.71 ab | 4.27 bcd |
| | | 25 | 600 c | 1575 f | 94.5 i | 150 de | 20.0 ef | 30.4 a | 12.0 a | 731 a | 2.54 bcd | 5.48 a |
| 50 | - | 0 | 695 b | 1381 g | 96.0 def | 175 b | 20.1 ef | 25.0 i | 9.0 i | 455 i | 2.76 a | 3.99 def |
| | NaCl | 30 | 527 fgh | 1826 ab | 96.3 cd | 132 f | 20.0 ef | 27.6 c–g | 10.0 gh | 553 fgh | 2.77 a | 3.67 fg |
| | | 50 | 535 efg | 1798 bcd | 95.9 ef | 137 f | 20.4 de | 27.1 fgh | 10.9 bcd | 594 cde | 2.48 d | 4.08 de |
| | CaCl$_2$ | 15 | 517 gh | 1863 ab | 96.2 cde | 132 f | 20.4 de | 27.8 c–f | 10.2 e–h | 568 efg | 2.74 a | 3.77 efg |
| | | 25 | 557 def | 1699 cde | 94.7 i | 165 c | 22.8 a | 28.3 bcd | 11.3 b | 641 b | 2.51 cd | 5.29 a |
| 20 | - | 0 | 800 a | 1211 h | 96.9 a | 190 a | 19.1 fg | 20.5 j | 7.9 j | 325 j | 2.59 a–d | 3.09 i |
| | NaCl | 30 | 530 gh | 1821 ab | 96.5 bc | 132 f | 19.9 ef | 26.3 h | 9.8 h | 520 h | 2.66 abc | 3.44 gh |
| | | 50 | 535 efg | 1808 abc | 96.7 ab | 122 g | 18.6 g | 26.6 gh | 10.1 e–h | 541 fgh | 2.62 a–d | 3.32 hi |
| | CaCl$_2$ | 15 | 517 gh | 1881 ab | 96.5 bc | 125 g | 19.5 efg | 27.2 e–h | 10.2 e–h | 556 fgh | 2.66 a–d | 3.48 gh |
| | | 25 | 562 de | 1698 de | 95.5 gh | 155 d | 21.6 bc | 27.5 d–g | 10.4 d–g | 575 def | 2.63 a–d | 4.46 bc |

### 3.5. Water Status, as Well as Chlorophyll and Carotenoid Contents

As the water deficit intensified, the RWC (a measure of the leaves' hydration level), chlorophyll content and carotenoid content shifted to lower values (Table 5). Independently of the water deficit severity, the pre-stress $CaCl_2$ applications generally enhanced the RWC, chlorophyll content and carotenoid content, especially at the highest concentration (25 mM). The pre-stress NaCl applications also promoted these traits, though this effect was significant under water deficit conditions (50 and 20% available water contents). For NaCl, the differences among the two concentrations (30 and 50 mM) were non-significant.

**Table 5.** Relative water content, electrolyte leakage, content of chlorophyll, carotenoids, malondialdehyde and proline as well as peroxidase and ascorbate peroxidase activity of periwinkle cv. 'Pacifica XP Really Red' plants initially receiving NaCl (0, 30 and 50 mM) or CaCl$_2$ (15 and 25 mM) via irrigation (five applications at 3 d intervals) and subsequently exposed to different watering levels (80, 50 and 20% available water content) during cultivation. At first application, plants were at the four-leaf stage. Means followed by different letters indicate significant differences (*p* = 0.05; comparison in columns). Values are the mean of four replications. FW, fresh weight, MDA, Malondialdehyde.

| Available Water Content (%) | Treatment | | Relative Water Content (%) | Chlorophyll a | Chlorophyll b | Chlorophyll (a + b) | Carotenoid | Electrolyte Leakage (%) | MDA | Proline | Catalase | Peroxidase |
| | Compound | Concentration (mM) | | Content (mg g$^{-1}$ FW) | | | | | Content (µmol g$^{-1}$ FW) | | Activity (µmol min$^{-1}$ g$^{-1}$ FW) | |
|---|---|---|---|---|---|---|---|---|---|---|---|---|
| 80 | - | 0 | 80.9 bcd | 6.0 cd | 3.2 ef | 9.2 ef | 2.2 cde | 23.7 f | 0.280 efg | 4.19 e | 0.001 j | 0.010 k |
| | NaCl | 30 | 81.7 bc | 6.4 bcd | 3.8 c | 10.3 cd | 2.5 bc | 23.1 f | 0.269 fg | 4.19 e | 0.007 i | 0.016 jk |
| | | 50 | 82.5 b | 6.4 bcd | 3.4 de | 9.8 cde | 2.2 cde | 23.6 f | 0.274 efg | 4.12 e | 0.008 h | 0.018 j |
| | CaCl$_2$ | 15 | 90.9 a | 8.6 a | 4.3 b | 13.0 b | 2.4 cd | 20.1 g | 0.238 gh | 4.09 e | 0.010 g | 0.014 jk |
| | | 25 | 93.8 a | 9.4 a | 5.0 a | 14.5 a | 3.1 a | 22.3 fg | 0.215 h | 3.34 e | 0.008 h | 0.015 jk |
| 50 | - | 0 | 66.6 f | 4.3 fg | 2.1 ij | 6.4 h | 1.5 f | 36.0 bcd | 0.395 b | 6.74 d | 0.012 f | 0.027 i |
| | NaCl | 30 | 77.6 cde | 6.9 bc | 2.4 hi | 9.3 def | 2.4 cd | 33.0 d | 0.311 def | 6.57 d | 0.013 e | 0.075 g |
| | | 50 | 77.2 cde | 6.3 bcd | 2.9 fg | 9.3 def | 2.4 cd | 34.2 cd | 0.312 def | 5.91 d | 0.014 d | 0.085 ef |
| | CaCl$_2$ | 15 | 76.2 de | 6.9 bc | 3.5 cde | 10.5 c | 2.5 c | 29.3 e | 0.299 def | 5.88 d | 0.016 b | 0.083 f |
| | | 25 | 81.6 bc | 7.0 b | 3.6 cd | 10.7 c | 2.9 ab | 28.2 e | 0.290 def | 5.83 d | 0.016 b | 0.092 e |
| 20 | - | 0 | 59.3 g | 3.5 g | 1.2 [1] | 4.7 i | 1.4 f | 43.0 a | 0.533 a | 16.85 a | 0.014 de | 0.065 h |
| | NaCl | 30 | 73.8 e | 4.9 ef | 1.8 jk | 6.7 h | 2.1 de | 36.5 bc | 0.380 bc | 12.95 b | 0.015 c | 0.114 d |
| | | 50 | 73.2 e | 4.8 ef | 1.6 k | 6.5 h | 2.0 e | 37.7 b | 0.386 b | 13.49 b | 0.015 b | 0.129 c |
| | CaCl$_2$ | 15 | 74.7 e | 5.6 de | 2.1 ij | 7.8 g | 2.1 de | 37.5 b | 0.335 cd | 12.57 b | 0.015 b | 0.160 b |
| | | 25 | 76.5 de | 6.0 cd | 2.8 gh | 8.8 f | 2.2 cde | 35.1 bcd | 0.322 de | 10.63 c | 0.017 a | 0.168 a |

### 3.6. Leaf Proline Content

As the water deficit intensified, a higher leaf proline level was noted in the leaves (Table 5). The spray treatments tended to decrease the leaves' proline content. At the most severe water deficit (20% available water content), the pre-stress applications decreased the leaves' proline content.

### 3.7. Electrolyte Leakage and Lipid Peroxidation

As the water deficit intensified, the leaves' electrolyte leakage and MDA content (a measure of lipid peroxidation) shifted to higher values (Table 5). Independently of the severity of the water deficit, the pre-stress $CaCl_2$ applications generally decreased the electrolyte leakage and MDA content. The pre-stress NaCl applications also decreased these features, though this effect was significant under water deficit conditions (50 and 20% available water contents). For NaCl, no significant difference was noted among the two concentrations (30 and 50 mM).

When pooling all the 15 treatments, the leaves' electrolyte leakage and MDA content were positively associated ($R^2$ of 0.59; see also Figure 2).

### 3.8. Enzymatic Activity

As the water deficit intensified, the activities of CAT and POD increased (Table 5). Independently of the water deficit severity, the pre-stress applications increased the CAT activity. The pre-stress applications also increased the POD activity, though this effect was significant under water deficit conditions (50 and 20% available water contents).

### 3.9. Principal Component Analysis

To identify and measure the components that control the influence across the treatments, a PCA was performed (Figure 4). The eigenvalues were assessed to ascertain the number of optimal principal components. The first two dimensions supported 63% of the total variation (Figure S4). The degree of a considerable impact of each trait reflected on the PCA was assessed by employing the cos 2 index (Figure S5). Amongst these descriptors, the internal $CO_2$ concentration, chlorophyll content, photosynthesis rate, leaf area and plant dry weight had a great imprint for the classification of individual units. The PCA, established on the first two components, disclosed the complicated associations between the treatments (Figure 4A). The first axis showed that the most substantial factor was established on the level of the water deficit of groups (NaCl and $CaCl_2$), as well as the concentration employed. The second axis supported the differentiation of discrete groups receiving pre-stress applications, though the shortest kinship was detected for plants under 20% of the available water content (without application) and plants under the control conditions (80% water content), receiving 25 mM of $CaCl_2$. This suggests that supplementing the above-mentioned moieties may deliver a 'rescue' phenotype for plants under water stress and may also act as a growth promotor. Moreover, negative and positive associations across the traits were apparent (Figure 4B). The majority of indices were found to be homogenous (implying a co-regulation), though some were found to be negatively correlated (e.g., the MDA level versus the total chlorophyll content or transpiration rate versus the electrolyte leakage).

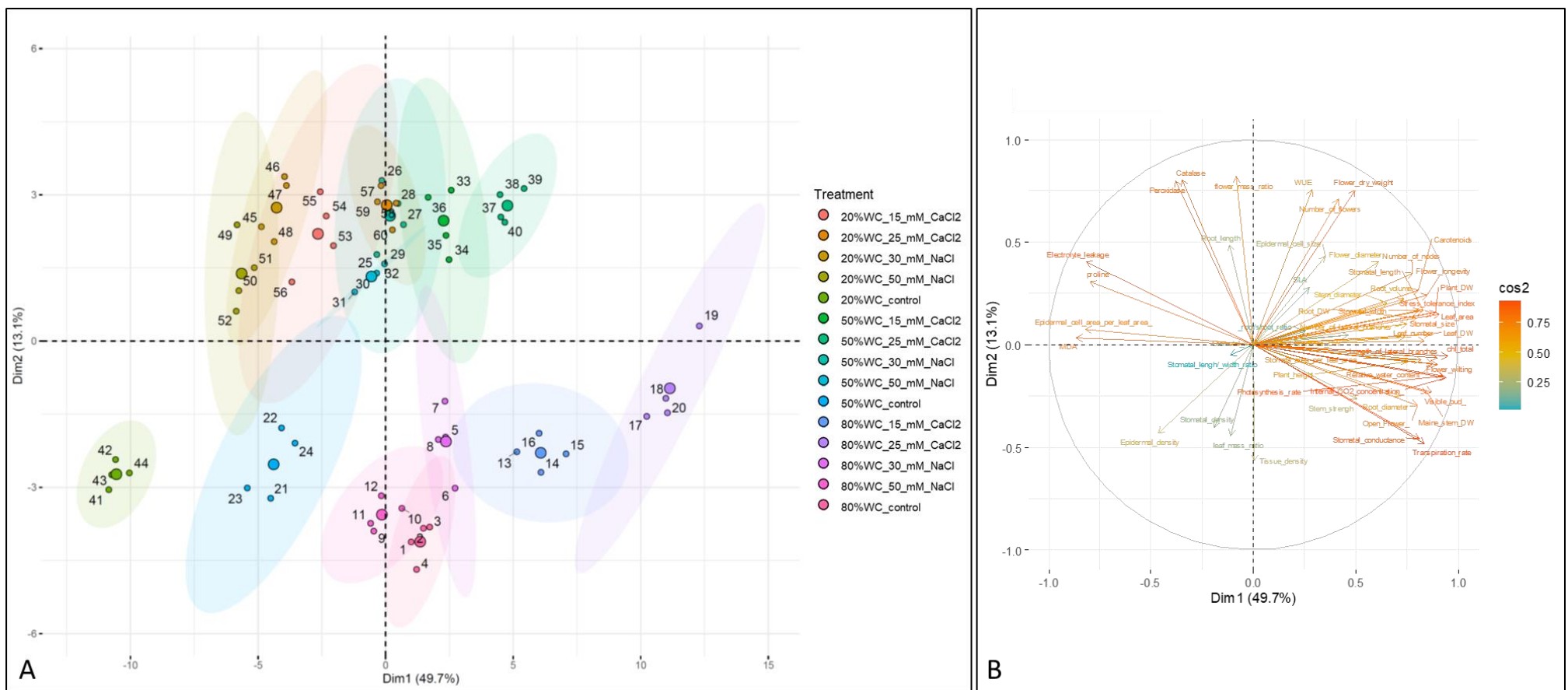

**Figure 4.** (**A**) Principal coordinate analysis across pre-stress applications [NaCl (0, 30 and 50 mM) or CaCl$_2$ (15 and 25 mM) via irrigation (five applications at 3 d intervals)] and subsequent watering levels (80, 50 and 20% available water content) in periwinkle cv. 'Pacifica XP Really Red'. Larger dots indicate mean values calculated from four discrete biological replications. (**B**) The contribution of each trait in the two dimensions is indicated by a gradient scale and color intensity (scale). Vectors near the plot center have lower cos2 values. Narrow angles among variables indicate affinity and wide angles a negative correlation.

## 4. Discussion

In the current investigation, the potential of using controlled osmotic stress (via NaCl or CaCl₂) to confer a cross-tolerance to the water deficit was evaluated by using a range of water severity conditions (80, 50 and 20% available water content) in periwinkle, which is commonly employed for different purposes including pot, garden or medicinal use [1,12]. To induce an equal increase in the irrigation water $Cl^-$ ionic content, $CaCl_2$ was added in half concentration of NaCl.

The ornamental quality includes several external (visible) and internal (non-visible) characteristics, determining the market price across the production–distribution chain [13]. The present results indicate that a water deficit adversely affected the critical external quality features, such as the flowers' size and plants' overall stature (main stem and lateral branch length; Table 2). Notably, the floral longevity was also strongly reduced under water deficit conditions (Table 1). Under these conditions, the stem strength also shifted to lower values (Table 2), indicating a higher vulnerability to bending [17]. These effects were collectively more prominent as the water deficit became stronger. The results of this study together with an earlier work [1] clearly illustrate that the water severity during cultivation negatively regulates the different aspects of the ornamental quality expanding from the external characteristics (the plant stature and flower size) and stem inclination possibility to the floral longevity, while these processes are up-regulated as the available water content of the growth medium becomes limited.

Under the control (non-stress) conditions (80% available water content), the application of NaCl adversely affected the flowers' size and overall plant stature (Table 2) but improved the floral longevity (Table 1). Under water deficit conditions, the pre-stress NaCl application exerted a positive effect on both the external quality characteristics and floral longevity, especially at the lowest employed concentration (30 mM). Independently of the water deficit severity, the pre-stress CaCl₂ application generally improved the above-noted morphological traits, especially at the highest employed concentration (25 mM). The pre-stress applications generally exerted limited effects on the stem bending sensitivity indices (Table 2). Therefore, the advantage of a NaCl application in terms of the ornamental quality is limited to the conditions of the water deficits severity, whereas a CaCl₂ application confers promotive effects which are independent of the water deficits severity. An application of CaCl₂ (1.5%) has also been proved beneficial for the transgenerational drought memory in wheat [29]. The foliar application of CaCl₂ (100 μM) also improved the post-stress recovery potential of *Camellia sinensis* [30], while seed priming with CaCl₂ ($\psi_s$ of $-1.25$ MPa; osmopriming, 100 mg $L^{-1}$, respectively) improved the chilling tolerance of wheat [9] and maize [31].

The intensity of the green color on the leaves commonly serves as an important indicator of a plant's health, vigor and aesthetic value at all steps of the production and distribution chain [13,32]. As the water deficit intensified, the chlorophyll content of the leaves shifted to lower values (Table 5). An adverse effect of the water's severity on the chlorophyll content of the leaves has also been earlier reported in periwinkle [1] and other species [3]. The pre-stress applications generally improved the chlorophyll content of the leaves, and this improvement was more pronounced as the water deficit intensified (Table 5). The pre-stress applications exerted a promotive effect on the content of the leaves under the control conditions, and they restored their value when the available water content of the growth medium decreased. Taken together, these results indicate that the water severity decreases the plants' aesthetic value by attenuating the intensity of the green color on the leaves, and this process is effectively mitigated by pre-stress CaCl₂ or NaCl applications.

Water deficit conditions were associated with smaller plants, an effect that was apparent across all organs (stem, leaves, flower and root) (Table 3). The plants cultivated under water deficit conditions intercepted less light (owing to the reduced leaf area) and showed a reduced photosynthetic performance (Figure 1). The latter was related to the attenuated leaf chlorophyll content and reduced stomatal conductance (thus a lower internal $CO_2$

concentration) (Figure 2). Pre-stress $CaCl_2$ or NaCl applications at least partly restored these contributing factors and alleviated the adverse effects of the water deficit on the plants' growth.

Environmental conditions during the plants' growth determined the anatomy of the stomatal features [27]. Having a lower available water content in the growth medium was associated with smaller stomata, which were denser (Table 4). This decrease in the size and increase in the density of stomata are typical effects of the water's severity and have been documented in periwinkle [1] and other species [15]. Pre-stress $CaCl_2$ or NaCl applications at least partly restored the above-noted effects of the water's severity. This effect of the pre-stress applications on the anatomy of stomatal features is most likely mediated by the advancement in the plants' hydration status (Table 5).

A water deficit was associated with an increased proline content in the leaves (Table 5). Under non-favorable conditions, proline facilitates the cell osmotic response and ROS detoxification [3,14]. Pre-stress $CaCl_2$ or NaCl applications induced a reduction in the proline content of the leaves, especially at the most severe water deficit (20% available water content). These results indicate that the positive effects of pre-stress $CaCl_2$ or NaCl applications on the plant's growth and performance are not through the proline metabolism-mediated pathway.

The water's severity was associated with a lower carotenoid content (antioxidant metabolite), but a higher activity of CAT and POD (antioxidant enzymes) (Table 5). An increase in the leaf electrolyte leakage and MDA content (a measure of lipid peroxidation) was also evident under water deficit conditions (Table 5), indicating the development of a cellular injury. Pre-stress $CaCl_2$ or NaCl applications upgraded the redox state by enhancing both the antioxidant metabolite (carotenoid) content and the activity of both the antioxidant enzymes under study (CAT and POD). Pre-stress $CaCl_2$ or NaCl applications further strongly alleviated the cellular injury symptoms (the lower electrolyte leakage, MDA content and chlorophyll degradation). Taken together, these results indicate that pre-stress $CaCl_2$ or NaCl applications promoted the plant redox state, effectively alleviated the related oxidative stress symptomatology and in this way conferred a tolerance to a water deficit.

Even though the landscape of priming and cross-tolerance seems to be multidimensional, while a range of agents may be used across different crop species, it appears that there is a common underlying mechanism delivering the tolerance. Stress factors often induce an abundance in ROS. Hence, an oxidation damage occurs, compromising the membrane's integrity and inducing an uncontrolled permeability. In the current study, it was established that a salt pre-treatment provided a memory effect and upregulated the antioxidant machinery of periwinkle plants during water stress, meaning that the priming of the plantlets may enhance a tolerance to subsequent stress events.

Compared to NaCl, the better efficiency of a $CaCl_2$ pre-stress application in conferring a cross-tolerance to a water deficit may be attributed to the positive role of Ca in both the cell membranes' stability and stress signaling. In one hand, Ca is involved in a range of cellular processes, including the membranes' structure [33,34]. For instance, Ca has been positively associated with both the membranes' thickness and rigidity [35]. In addition, the cells' stability is maintained by the interactions between Ca and phospholipids, being the membrane components [36]. On the other hand, it is well documented that Ca is a crucial second messenger acting as a mediator in regulating and specifying the cellular responses to environmental stresses [37,38]. Abiotic stress frequently elicits increases in the cytosolic free Ca and activates the Ca-dependent signaling pathway, stimulating the appropriate response [39].

The application of agents other than $CaCl_2$ and NaCl has also been used in order to provide priming effects against abiotic stresses across crops. $KNO_3$ (2.5 and 5%) and $SiO_2$ (3 and 3.5%) were recently used in order to improve the emergence and seedling growth in rice under drought conditions by upregulating the antioxidant enzyme capacity [40]. The protective effects of $KNO_3$ as well as of urea were also established in several maize genotypes [41]. Treatment with $ZnSO_4$ and $Mg(NO_3)_2$ salts also triggered antioxidant

and physio-biochemical responses, and provided a water stress adaptation in wheat [42]. Moreover, a silicon application parallel to foliar spraying with sulphur salts was shown to confer a drought tolerance in corn [43].

## 5. Conclusions

The possibility of using NaCl or $CaCl_2$ applications to confer a cross-tolerance to a water deficit was investigated in periwinkle. A cultivation under a limited water supply degraded both the plants' aesthetic value (the plants' stature, flower size and leaf coloration) and floral longevity. The reduced (root and shoot) biomass accumulation was associated with a smaller light capture (via a reduced leaf area) and a lower photosynthetic performance (via a reduced chlorophyll content and stomatal conductance). A growth impairment was also related to the advancement in oxidative stress-induced symptoms (chlorophyll and membrane degradation, as well as lipid peroxidation). The above-mentioned effects were more prominent as the water deficit intensified. Pre-stress $CaCl_2$ or NaCl applications generally restored the water severity-induced effects. $CaCl_2$ was generally more effective in inducing these promotive effects compared to NaCl. For $CaCl_2$, the highest concentration (25 mM) was generally optimal, whereas NaCl was the lowest one (30 mM). Under the control (non-stress) conditions, $CaCl_2$ was consistently associated with positive effects, whereas some negative effects were induced by NaCl. Taken together, these results indicate that pre-stress $CaCl_2$ or NaCl applications confer a cross-tolerance to a water deficit and promote the aesthetic value and floral longevity, especially under a severe water deficit.

**Supplementary Materials:** The following supporting information can be downloaded at: https://www.mdpi.com/article/10.3390/horticulturae8111091/s1. Figure S1. Plant dry mass of periwinkle cv. 'Pacifica XP Really Red' plants receiving different concentrations of $CaCl_2$ (A) or NaCl (B) via irrigation (five applications at 3 d intervals). At first application, plants were at the four-leaf stage. Dry mass measurements were carried out 5 d after the last salt application. Values are the mean of four replications ± SE. Figure S2: Representative images of periwinkle 'Pacifica XP Really Red' plants initially receiving $CaCl_2$ (0, 15 and 25 mM corresponding to left, middle and right plant in each panel, respectively) via irrigation and subsequently exposed to different watering levels (80, 50 and 20% available water content corresponding to top, middle and bottom panel, respectively) during cultivation. Figure S3: Representative images of periwinkle 'Pacifica XP Really Red' plants initially receiving NaCl (0, 25 and 50 mM corresponding to left, middle and right plant in each panel, respectively) via irrigation, and subsequently exposed to different watering levels (80, 50 and 20% available water content corresponding to top, middle and bottom panel, respectively) during cultivation. Figure S4: The first ten principal components and percentages of attributed variation. The first two eigenvalues were used to construct the principal component analysis biplot (accounting for 62.8% of the cumulative percentage explained); Figure S5: Quality of representation (cos2) of the variables on factor map. Variables on the first five dimensions are displayed. Size and color intensity correlates to a better representation of specific traits.

**Author Contributions:** Conceptualization, A.R.N. and D.F.; methodology, N.Z., A.R.N., S.M.-F., H.F. and D.F.; software, N.N. and D.F.; validation, D.F.; formal analysis, N.N. and D.F.; resources, A.R.N. and S.M.-F.; data curation, N.Z., A.R.N., S.M.-F., N.N. and D.F.; writing—original draft preparation, N.Z. and D.F.; writing—review and editing, N.N., A.R.N. and D.F.; supervision, A.R.N. and D.F.; project administration and funding acquisition, A.R.N. and S.M.-F. All authors have read and agreed to the published version of the manuscript.

**Funding:** This research was financed by Lorestan University (Iran).

**Institutional Review Board Statement:** Not applicable.

**Informed Consent Statement:** Not applicable.

**Data Availability Statement:** Raw data are available upon request from the corresponding author.

**Acknowledgments:** The authors gratefully acknowledge the laboratory crew for their inputs, continued attentiveness and lifelong dedication to service. The valuable comments of the editor and three anonymous reviewers are greatly appreciated.



**Conflicts of Interest:** All authors declare no conflicts of interest.

**Abbreviations**

CAT     Catalase
MDA     Malondialdehyde
PCA     Principal component analysis
POD     Peroxidase
ROS     Reactive oxygen species
RWC     Relative water content

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
