# Peer review of "Efficiency of Sodium and Calcium Chloride in Conferring Cross-Tolerance to Water Deficit in Periwinkle"

_horticulturae, doi:10.3390/horticulturae8111091_

Round 1

Reviewer 1 Report

In this manuscript the authors pre-treated plants of periwinkle with sodium and calcium chloride to find out if these treatments may induce cross-tolerance to later water deficit.

The subject is in the scope of Horticulturae and is of some scientific interest.

Although I understand the scientific importance of this study, I have some questions and suggestions for authors consideration,  and better address this matter and improve manuscript preparation.

Line 46 and 52: I suggest changing "avenues" to "ways" and "immerse" to "great"

Line 63:  While there are many studies to find ways to circumvent the effect of a saline environment on plant growth, authors are proposing to apply salt in the plant environment (to reduce the damage of future drought stress). These salt applications will induce later problems by soil salinization? In my opinion, pre-stress treatments could be interesting to be applied (for example) only on nursery plants, then leach out salts from the substrate just before moving plants to definitive soil where they will be grown (and probably be subjected to drought stress). I mentioned soil because I think that If plants will not be planted under field conditions (where frequently water stress occur), but will be conducted under greenhouse (and irrigated) conditions, then there are no expectations of future drought stress occurrence and, consequently, no need of pre-stress treatments.

So, I think authors should better address and discuss these (above-mentioned) questions in the manuscript introduction as well as in the discussion.

Lines 67 -73: I suggest leaving this information to material and methods.

Line 99-100: It is necessary to give more detailed information on how these pre-treatments were implemented. For example, these salt solutions were applied in what amount in each of the five pre-stress applications? They were applied in an amount enough to substrate saturation? Or in an amount enough to reach substrate field capacity? Also, what was the total amount (summing all five applications) of salts that were applied (e.g. mols per liter of growth media)?

After the five pre-stress applications, did all the salts applied was remained in the growth media, or were leached out from the growth media just before drought stress treatments?

So, if NaCl and CaCl2 remained in the growth media, it seems to me that it should not be considered only as "pre"-stress, once applied salts certainly acted as simultaneous stress when water stress treatments were imposed. Under this situation, plants were subjected to salt stress + water stress.  Because of that I have suggested authors to better address this subject in the introduction, and better discuss these situations in the result discussion.

Author Response

Reviewer #1: In this manuscript the authors pre-treated plants of periwinkle with sodium and calcium chloride to find out if these treatments may induce cross-tolerance to later water deficit.

The subject is in the scope of Horticulturae and is of some scientific interest.

Although I understand the scientific importance of this study, I have some questions and suggestions for authors’ consideration, and better address this matter and improve manuscript preparation.

Line 46 and 52: I suggest changing "avenues" to "ways" and "immerse" to "great"

Authors: Done (Lines 45, 51).

Line 63:  While there are many studies to find ways to circumvent the effect of a saline environment on plant growth, authors are proposing to apply salt in the plant environment (to reduce the damage of future drought stress). These salt applications will induce later problems by soil salinization? In my opinion, pre-stress treatments could be interesting to be applied (for example) only on nursery plants, then leach out salts from the substrate just before moving plants to definitive soil where they will be grown (and probably be subjected to drought stress). I mentioned soil because I think that If plants will not be planted under field conditions (where frequently water stress occur), but will be conducted under greenhouse (and irrigated) conditions, then there are no expectations of future drought stress occurrence and, consequently, no need of pre-stress treatments.

So, I think authors should better address and discuss these (above-mentioned) questions in the manuscript introduction as well as in the discussion.

Authors: In principle, we agree with the reviewer, and share the raised concerns. However, the scope of the study is not to advise the application of salts in crop fields uncritically. That would create an increment of ion concentration which could have an adverse effect in future cultures (as suggested). We aim to provide data on cross-tolerance across abiotic stresses (salinity to drought), as well as to ascertain the type of ions that would have a beneficial role, along with the optimal dosage. Ideally, treatment should occur as the reviewer proposes in nurseries (thus before transplantation). Periwinkle cultivation may be continued in greenhouses, hydroponics (though that does not automatically translate to ample water availability/or optimal water quality), or even open fields; since the plant is also used as a source of bioactive metabolites. In the current study, it was also established that exposure of periwinkle plants to salinity can provide similar phenotypes to the control group, and therefore an improved water economy can be achieved.

Lines 67–73: I suggest leaving this information to material and methods.

Authors: Done (Lines 69–70).

Line 99–100: It is necessary to give more detailed information on how these pre-treatments were implemented. For example, these salt solutions were applied in what amount in each of the five pre-stress applications? They were applied in an amount enough to substrate saturation? Or in an amount enough to reach substrate field capacity? Also, what was the total amount (summing all five applications) of salts that were applied (e.g. mols per liter of growth media)?

After the five pre-stress applications, did all the salts applied was remained in the growth media, or were leached out from the growth media just before drought stress treatments?

So, if NaCl and CaCl2 remained in the growth media, it seems to me that it should not be considered only as "pre"-stress, once applied salts certainly acted as simultaneous stress when water stress treatments were imposed. Under this situation, plants were subjected to salt stress + water stress.  Because of that I have suggested authors to better address this subject in the introduction, and better discuss these situations in the result discussion.

Authors: This is now dealt with in the Materials and Methods (Lines 102–104). Following pre-stress applications, the applied salts were leashed out by dual irrigation with double-distilled water. 

Reviewer 2 Report

It is a very interesting scientific problem to explore the drought adaptability of plants under climate change. In this study, periwinkle was taken as the
research object, and a large number of phenotypic and physiological indicators were measured through pre-stress and drought stress. However, there are many needs for research areas to be improved:

1. The experimental design was not perfect, so it was suggested that the pre-stress and drought stress treatment should adopt orthogonal design to screen the optimal concentration;

2. The introduction lacks the description of the research object, especially the reason why periwinkle was selected for drought research. In addition, the
current progress of pre-stress has not been explained clearly;

3. The presentation method of the obtained data is inappropriate. For example, half of the picture in Figure 1 already contains all the results. It is
recommended to delete the duplicate content;

4. The comparison with similar research results of other species should be added in the discussion part;

5. Table 2 - Table 6 is seriously homogenized. It is suggested that several of them can be made into pictures;

6. The typical pictures need to be added, including the morphological comparison of different stages, the structural changes of stomata and epidermal cells of periwinkle after treatment.

To sum up, the current form is not recommended to be published, and it is recommended to carefully modify and then resubmit.

Author Response

Reviewer #2: It is a very interesting scientific problem to explore the drought adaptability of plants under climate change. In this study, periwinkle was taken as the research object, and a large number of phenotypic and physiological indicators were measured through pre-stress and drought stress. However, there are many needs for research areas to be improved:

1. The experimental design was not perfect, so it was suggested that the pre-stress and drought stress treatment should adopt orthogonal design to screen the optimal concentration;

Authors: A large preliminary study was employed, where appropriate concentrations were selected. Then, 15 experimental units (5 pre-stress treatments × 3 water deficit levels) were realized as a factorial experiment. Owing to the involved labor, more experimental units could not be accommodated.    

2. The introduction lacks the description of the research object, especially the reason why periwinkle was selected for drought research. In addition, the current progress of pre-stress has not been explained clearly;

Authors: This is now mentioned in the first section of the Materials and Methods (Lines 69–70).

3. The presentation method of the obtained data is inappropriate. For example, half of the picture in Figure 1 already contains all the results. It is
recommended to delete the duplicate content;

Authors: As suggested, Figure 1 is now corrected (newly numbered Fig. 2).

4. The comparison with similar research results of other species should be added in the discussion part;
Authors: This is now provided in the discussion (Lines 490–494).

5. Table 2–Table 6 is seriously homogenized. It is suggested that several of them can be made into pictures;
Authors: As suggested, selected core parameters are now depicted in Figures (newly added Figs. 1 & 3).

6. The typical pictures need to be added, including the morphological comparison of different stages, the structural changes of stomata and epidermal cells of periwinkle after treatment.
Authors: As suggested, representative pictures are now provided (newly added Figs. S1 & S2).

To sum up, the current form is not recommended to be published, and it is recommended to carefully modify and then resubmit.

Reviewer 3 Report

The article considers the efficiency of treatment of periwinkle [Catharanthus roseus (L.) G. Don] plants with NaCl or CaCl2 solutions in two concentrations on mitigation of drought stress in plants. A decrease in the severity of drought stress was shown by plant appearance and biochemical parameters. Higher efficacy was found for 25 mM CaCl2.

The article is written in a good language, the experimental design and methods are described in detail, the results are correctly processed statistically and presented in detailed tables.

However, there are several questions to the article.

1. In the title and throughout the text of the article the authors write "cross tolerance", although the work studied the effect of only one stressor, namely water deficit. Therefore, it should be written exactly about resistance to water stress.

2. The choice of salts and their concentrations to assess the effect on drought stress tolerance is very poorly justified. For example, according to the data of the work, the best effect was obtained in periwinkle plants at the highest concentration of CaCl2 (25 mM). But if you increase this dose, perhaps the effect will be more significant.  

3. Increasing the resistance of plants to various stresses through treatment with salts, including CaCl2, has been known for over 70 years. Hundreds and thousands of articles have been published, and the manuscript also contains references to a number of works on increasing plant resistance to water deficit or drought by treatment with CaCl2 solutions. This gives the question: what new did the authors want to say with their article - just a comparison of NaCl and CaCl2, and their concentrations? The authors should redo the hypothesis, purpose and objectives of the study so that the good results obtained in the study received a serious scientific treatment.  Maybe make additional comparisons with other salts (with Mg, K, SO4, NO3 or other ions), with other stresses to make a convincing case for cross-tolerance, etc.

4. Page 3, lines 116-118. It is rather doubtful that during the 11 weeks of plants growing in the greenhouse under natural conditions, the average temperature was 22.2±1.6 °C. Even during 24-hour day it varies in much higher dimensions.

Author Response

Reviewer #3: The article considers the efficiency of treatment of periwinkle [Catharanthus roseus (L.) G. Don] plants with NaCl or CaCl2 solutions in two concentrations on mitigation of drought stress in plants. A decrease in the severity of drought stress was shown by plant appearance and biochemical parameters. Higher efficacy was found for 25 mM CaCl2.

The article is written in a good language, the experimental design and methods are described in detail, the results are correctly processed statistically and presented in detailed tables.

However, there are several questions to the article.

  1. In the title and throughout the text of the article the authors write "cross tolerance", although the work studied the effect of only one stressor, namely water deficit. Therefore, it should be written exactly about resistance to water stress.

Authors: It is now clarified that we refer to cross tolerance to water stress in several passages throughout the text.

  1. The choice of salts and their concentrations to assess the effect on drought stress tolerance is very poorly justified. For example, according to the data of the work, the best effect was obtained in periwinkle plants at the highest concentration of CaCl2 (25 mM). But if you increase this dose, perhaps the effect will be more significant. 

Authors: Fifteen factors (5 pre-stress treatments × 3 water deficit levels) were realized in the current study. More concentrations were studied in the preliminary study, where the optimal concentration range was selected.

  1. Increasing the resistance of plants to various stresses through treatment with salts, including CaCl2, has been known for over 70 years. Hundreds and thousands of articles have been published, and the manuscript also contains references to a number of works on increasing plant resistance to water deficit or drought by treatment with CaCl2 solutions. This gives the question: what new did the authors want to say with their article - just a comparison of NaCl and CaCl2, and their concentrations? The authors should redo the hypothesis, purpose and objectives of the study so that the good results obtained in the study received a serious scientific treatment.  Maybe make additional comparisons with other salts (with Mg, K, SO4, NO3 or other ions), with other stresses to make a convincing case for cross-tolerance, etc.

Authors: This is now provided in the discussion (Lines 540–547 & 559–567).

  1. Page 3, lines 116–118. It is rather doubtful that during the 11 weeks of plants growing in the greenhouse under natural conditions, the average temperature was 22.2±1.6 °C. Even during 24-hour day it varies in much higher dimensions.
    Authors: In the manuscript, the mean value is depicted. As suggested, the range of the respective values is now provided (Lines 118–120).

Round 2

Reviewer 2 Report

The author has finished the modification and I agree to accept it.

Author Response

Reviewer #2: The author has finished the modification and I agree to accept it.

Authors: Thank you.

Reviewer 3 Report

The authors have taken most of the reviewer's comments into account and have made appropriate changes in the text of the revised manuscript. However, there are still some questions and comments on the manuscript. 

1. Regarding the commentary on the term "cross-adaptation," which is used throughout the article. The authors gave an explanation on p. 3, lines 93-95. I believe this phrase should be moved to the Abstract and Introduction to clarify the use of the term in more detail. 

2. Comment 2 about the choice of concentrations of the salts used in the experiment was ignored by the authors. Maybe some results of preliminary experiments with different concentrations should be included in the article - in the main text or supplemental material?

3. The Introduction should be more specific about the novelty of the data presented in the article, due to the numerous publications previously.  

I believe that the inclusion of additional information in the text of the article will improve the accuracy of the presentation and the understanding of the content by the readers. 

Author Response

Reviewer #3: The authors have taken most of the reviewer's comments into account and have made appropriate changes in the text of the revised manuscript. However, there are still some questions and comments on the manuscript. 

1. Regarding the commentary on the term "cross-adaptation," which is used throughout the article. The authors gave an explanation on p. 3, lines 93-95. I believe this phrase should be moved to the Abstract and Introduction to clarify the use of the term in more detail. 

Authors: The term ‘cross-stress tolerance’ is presented in the Title (Line 3), and Abstract (Lines 19 and 33), while it is explained in the Introduction (Lines 46–51).

2. Comment 2 about the choice of concentrations of the salts used in the experiment was ignored by the authors. Maybe some results of preliminary experiments with different concentrations should be included in the article - in the main text or supplemental material?

Authors: It is now added as new Figure S1. Concentrations of  15 and 25 mM CaCl2 have been chosen based on their positive effects on plant dry mass, and corresponding concentrations of NaCl (equal EC level and Cl- content) have been selected.  

3. The Introduction should be more specific about the novelty of the data presented in the article, due to the numerous publications previously.

Authors: It is now clarified in the Introduction (Lines 62–66).  

I believe that the inclusion of additional information in the text of the article will improve the accuracy of the presentation and the understanding of the content by the readers.